# Sequence-Dependent Nanofiber Structures of Phenylalanine and Isoleucine Tripeptides

**DOI:** 10.3390/ijms21228431

**Published:** 2020-11-10

**Authors:** Qinsi Xiong, Ziye Liu, Wei Han

**Affiliations:** State Key Laboratory of Chemical Oncogenomics, School of Chemical Biology and Biotechnology, Peking University Shenzhen Graduate School, Shenzhen 518055, China; 1401111239@pku.edu.cn (Q.X.); ziyeliu@pku.edu.cn (Z.L.)

**Keywords:** peptide self-assembly, sequence–structure relationship, hybrid-resolution force field, molecular dynamics simulation, conformational dependence

## Abstract

The molecular design of short peptides to achieve a tailor-made functional architecture has attracted attention during the past decade but remains challenging as a result of insufficient understanding of the relationship between peptide sequence and assembled supramolecular structures. We report a hybrid-resolution model to computationally explore the sequence–structure relationship of self-assembly for tripeptides containing only phenylalanine and isoleucine. We found that all these tripeptides have a tendency to assemble into nanofibers composed of laterally associated filaments. Molecular arrangements within the assemblies are diverse and vary depending on the sequences. This structural diversity originates from (1) distinct conformations of peptide building blocks that lead to different surface geometries of the filaments and (2) unique sidechain arrangements at the filament interfaces for each sequence. Many conformations are available for tripeptides in solution, but only an extended β-strand and another resembling a right-handed turn are observed in assemblies. It was found that the sequence dependence of these conformations and the packing of resulting filaments are determined by multiple competing noncovalent forces, with hydrophobic interactions involving Phe being particularly important. The sequence pattern for each type of assembly conformation and packing has been identified. These results highlight the importance of the interplay between conformation, molecular packing, and sequences for determining detailed nanostructures of peptides and provide a detailed insight to support a more precise design of peptide-based nanomaterials.

## 1. Introduction

Supramolecular design has exhibited great potential in the field of material chemistry [1]. Self-assembled peptide materials are the most widely explored supramolecular biomaterials due to their high chemical versatility achieved from a combination of 20 amino acids in the peptide building process [2,3,4]. Such diversity exists even in very short peptides, such as those pioneered by Gazit [2,3], which consist of only two or three amino acids and give rise to discrete assembled nanomaterials with a wide range of applications in the fields of optoelectronic materials and biomedical materials [3,5,6,7].

A major challenge in the design of self-assembling short peptides is the establishment of the relationship between the supramolecular nanostructures and chemical structures of the building blocks [8,9]. Self-assembly into targeted structures involves a precisely controlled formation of various types of noncovalent interactions, such as π–π stacking and hydrogen bonding interactions [10]. Peptides capable of this behavior are believed to adopt special backbone conformations with a proper arrangement of a set of amino acid sidechains. Thus, the self-assembly ability of peptides is encoded in their amino acid sequences but in a manner that remains poorly understood.

Much experimental effort has been devoted to deciphering the relationship between the sequence of peptide building blocks and the resulting nanostructures [11,12,13,14,15]. With dipeptides or tripeptides as model systems, numerous studies have shown that assembled structures are highly dependent on sequence [13,16]. Even peptides with the same amino acid composition exhibit a markedly different assembly behavior if the sequence of the amino acids is changed [16]. The resulting self-assembled structures exhibit various modes of molecular packing and secondary structures, such as β-sheets, α-helices, and ϒ-turn conformations [17,18,19,20,21,22]. Several simple rules have been derived to rationalize the observed sequence–structure dependence [9,23], and attempts have been made to relate this dependence to the impact of sequence on the intrinsic conformational preference of monomeric peptides [3,15,16,24,25,26,27,28,29]. The molecular basis of the connection between peptide sequence and assembled structures remains obscure, however, largely due to the limitations of experimental characterization techniques [8].

Molecular dynamics simulations are useful for studying the self-assembly behavior of short peptides. Depending on the model representation, mechanistic insights with various levels of detail can be obtained [4,23,26,30,31]. All-atom models can be used to explore the full atomic details of the molecular organization and conformation in assembled structures, but they are limited to the simulation of relatively small systems and short timescales. As a more practical solution, coarse-grained (CG) models simplify the spatial representation, boost simulation efficiency, and have been used to examine the self-assembly of short peptides. This type of model has been used to explore the entire sequence space of dipeptides or tripeptides in a search for candidates with a high tendency to aggregate [23,32], and a set of empirical rules for the prediction of aggregation tendency has been developed. Practical as they are, these CG models cannot address a fundamental question regarding the precise roles of sidechain interactions and backbone conformations in the control of the assembly behaviors of peptides and the resulting assembly structures due to a lack of a detailed description of peptide molecules.

In our previous work, we developed a hybrid-resolution model termed PACE in which the solvent model is coarse-grained for simulation efficiency, while the atomic details of peptides are largely retained [33]. Originally developed for the study of protein folding, this model has recently been extended to the study of the self-assembly of phenylalanine dipeptides [34]. The model is able to not only describe reasonably well different types of noncovalent interactions, but also provide a balanced description of various conformations that the peptide can assume in both monomeric and assembled states. With this model, the dependence of assembled structures on the conformations of phenylalanine dipeptide has been established.

In this study, we employed the PACE model to explore the sequence–structure relationship of self-assembled peptides. The system focused here is a terminally end-capped tripeptide model containing two types of hydrophobic amino acids, namely, isoleucine (I) and phenylalanine (F), that have been frequently investigated in the studies of peptide-assembled nanostructures [9,14,15,26,29] (Figure 1A). We simulated the assembly behavior of tripeptides with all possible combinations of these amino acid types and found that the assembled structures are sensitive to sequence, which in general agrees with experimental results. Although the change of sequence affects the conformational preference of peptides in monomeric states, little correlation was found between the conformations of peptides in monomeric and assembled states, highlighting the difficulty in predicting assembly structures based only on monomeric conformational preferences. Intriguingly, only a few conformations were observed in the assembled states, and each exhibited a unique spatial arrangement of the amino acid sidechains on both sides of the backbone. With each sequence, the peptide was allowed to assume only specific assembly conformations so as to attain the best overall noncovalent packing in the assembled state. These results reveal the complex interplay between sequence, conformation, and noncovalent interactions in the determination of the final assembly structures. There is a need to understand this interplay at the assembly level to derive empirical rules for the design of peptide-based nanomaterials.

## 2. Results

### 2.1. General Tendency of Model Tripeptides to Assemble into Fibrous Structures

A total of eight tripeptides with a unique sequence containing isoleucine (I) and phenylalanine (F) were investigated in the present study. Each tripeptide is denoted by a three-letter code throughout this paper (Figure 1A). For each sequence, three self-assembly simulations (see Methods and Appendix A), each with a length of ~3 μs, were conducted independently with the improved PACE [34]. Each simulation began with 30 tripeptides randomly distributed in a box with a size of 6 × 6 × 6 nm^3^ at ~100 mg/mL, a concentration higher than that used in experimental studies but comparable to those in previous computational studies of peptide self-assembly. We observed the formation of fibrous aggregates usually within the first half of a microsecond of the simulations (Figure 1B), and the aggregates, once formed, remained stable till the end of the simulations. These fibers were usually 2–2.5 nm in diameter most often with a left-handed twist. The resulting aggregates were separated from their periodic images under periodic boundary conditions (PBCs), ensuring that the observed tendency to form fibers was not artificial, due to the PBCs.

Previous studies have shown that many other tripeptides can assemble into nanostructures with various types of morphologies [9] and that their ability to self-assemble into nanostructures is highly dependent on their amino acid sequence [13,14,15,16]. Consequently, it was intriguing to observe in our simulations the formation of fiber-like structures for all of the eight model tripeptides that were examined. To further validate this computational result, we experimentally characterized the assembly structures of these peptides. A sample of each peptide was prepared at 2.0 mg/mL under an annealing condition to avoid the formation of kinetically trapped structures (see Methods). The scanning electron microscopy (SEM) images of these samples showed that all the eight peptides indeed gave rise to fibrous assemblies, albeit with variations in the thickness and stiffness of these fibrous assemblies (Figure 1B). The fiber-forming propensity is therefore a shared property of tripeptides containing only Phe or Ile. That the assembly behaviors of all the eight tripeptides were correctly predicted indicates the robustness of our computational approach.

### 2.2. Sequence Dependence of Peptide Conformations in Monomers and Assemblies

Although assemblies with similar fibrous morphologies were observed for all the tripeptides examined, visual inspection of the internal structures of these assemblies revealed apparent differences between the sequences. We analyzed the conformational preference in the assemblies formed by different tripeptides to examine how the sequence of a tripeptide could affect the way it folded in its assembled states. For comparison, the conformational preference of each tripeptide in its monomeric states was also examined through a 2.4 μs replica exchange molecular dynamics (REMD) simulation of a single copy of this tripeptide in water (see Methods). Following our previous work, we categorized the tripeptide conformations sampled into a set of backbone rotamer states (see previous work [34] and the Appendix A). Each conformational state of a tripeptide was categorized based on the backbone conformational states of the three amino acid residues, each denoted by a three-letter code (Appendix A). Each letter represented one of four possible conformational states of a single residue, including right- (α_R_) and left-handed (α_L_) helical conformers, extended beta-conformer (β), and left-handed polyproline conformer (PP_II_) (Appendix A). With this clustering scheme, as many as 48 conformations can be defined uniquely.

This clustering analysis suggests that the monomeric tripeptides exhibit diverse conformations with no single cluster accounting for >25% of the population (Figure 1C and Appendix A), which was expected for short peptides. Nevertheless, the conformational preference of these tripeptides indeed varied with their sequences, as indicated by a comparison of the representative structures of the most populated conformations between the peptides (Figure 1C). In contrast, the conformations of the peptides in the assembled states are less heterogeneous. The most populated assembled conformations normally account for 40–75% of the population with the exception of IIF (Figure 1D and Appendix A). In each case, the most favored conformation in solution usually shares little similarity with that in the assemblies (Appendix A). It seems unlikely that one could deduce the assembly conformations of tripeptides based only on information about their monomeric conformational preference.

### 2.3. Geometric Features of Assembled Conformations of Tripeptides

To further gain insight into the origin of the conformational preference of the peptides in the assembled states, we analyzed the rotation angles of the sidechains, defined as the dihedral angle (*θ*) between the *C_α_*–*C_β_* vectors of adjacent residues. This order parameter can be used to describe the relative orientations of sidechains and thus the overall geometries of the peptide in a given conformational state. When *θ* is large (>115°), the two adjacent sidechains are arranged on the opposite sides of the backbone in an *anti* arrangement; when *θ* is small (<60°), the two sidechains are on the same side in a *syn* arrangement. Our previous study showed that this rotation angle of sidechains is a key geometric factor determining the morphologies of the assemblies formed by diphenylalanine [34]. For tripeptides, two rotation angles (*θ*_1_ and *θ*_2_) are needed to describe the relative orientations of all three sidechains.

Figure 2 shows the typical orientations of sidechains for all the 48 possible conformational states of a tripeptide. On the basis of *θ*_1_ and *θ*_2_, these conformations can be grouped into five categories. For those in group I, called here all-*anti* conformation, any two adjacent sidechains are always on the opposite sides of the backbone; for group II (called all-*syn*), all the sidechains are on the same side; and the conformations of groups III and IV (termed N-*syn* and C-*syn*, respectively) have either the N-terminal or C-terminal sidechain pointing in the same direction as the sidechain of the middle residue, while the residue at the other terminus points in the opposite direction. The conformations of the last group (shown in the shaded region of Figure 2) have at least a pair of adjacent sidechains that do not point in either the same direction or the opposite directions.

To explore the conformational space accessible to the peptides in the assemblies, we examined which of the 48 conformational states were observed with a non-negligible probability (>5%) in the assembled states for at least one of the sequences (indicated by small triangles in Figure 2). The result suggests that none of the conformations from the last group could be sampled in the assembly simulations for any sequence. Thus, any conformation involving a sidechain arrangement that was neither *syn* nor *anti* appeared to be incompatible with the assembled structures. For each of the first four groups, at least one of its members was observed in the assembled states, indicating that all four types with the aforementioned sidechain orientation were probably accessible in the assembled states of tripeptides. However, not every member of these groups was sampled. Of all the accessible conformations, *βββ* from group I (all-*anti*) and *α_R_α_R_β* from group II (all-*syn*) were the two predominant conformations observed in the assemblies. *βββ* is the most favorable assembled conformation for FFF, FIF, and III, while *α_R_α_R_β* is the most frequently observed for the other sequences.

To further validate the significance of the two conformations, we compared them with seven other assembled tripeptides whose assembly structures have been resolved to date (Table 1). Of the seven peptides, DYF [16] and Ac-YLD [35] were found to adopt *βββ*, and Ac-LLE [36] was found to adopt *α_R_α_R_β*. Although the remaining three peptides were found to adopt other conformations, this “abnormality” can be rationalized as these peptides either contained a noncoded amino acid at the N-terminus that strongly favors *α_R_* as in Boc-Leu-Val-Ac12c-OMe [17], or as in Pro/Hyp-FF [21] contained a proline or a hydroxyproline at the C-terminus that has a rather different conformational distribution from normal α-substituted amino acids, or was locked into *α*_L_*α*_L_PII via the formation of a strong internal salt bridge between its sidechain and the uncapped N-terminus as in YFD [16]. Given their preference for various tripeptide sequences, the two conformation states appear to be generic conformations that tripeptides could adopt to form fibrous assemblies.

As a whole, our analysis suggests that there is only a small fraction of conformational space in which tripeptides are allowed to self-assemble into fibrous assemblies. However, there is a large difference in sidechain arrangement between the allowed conformations, and the preference for these conformations is sensitive to the peptide sequences.

### 2.4. Sequence Dependence of Molecular Packing of Tripeptides in Fibrous Assemblies

Having examined the conformational features of the tripeptides, we next analyzed how the peptides were organized in the fibrous assemblies. As shown in Figure 1B, despite the difference in sequence and conformation, most of the tripeptides with the exception of IIF were found to assemble into fibrous structures made of three or four filaments. In each filament, the peptides stacked with one another through in-register hydrogen bonding (HB) interactions in the direction of the fiber axes. This type of in-register parallel alignment of peptide chains has most commonly been seen in the crystal structures of tripeptide assemblies reported to date [21,35]. The distance between adjacent peptide chains in a filament ranges between 4.6 and 4.9 Å, which is also consistent with a typical XRD signal of β-strand aggregates (4.9 Å) [14]. These filaments were further intertwined with each other through hydrophobic interactions between their lateral surfaces.

Figure 3A illustrates the details of molecular packing between filaments with various tripeptide sequences. When the peptides were composed of only phenylalanine, about 88% of them were obliged to mainly fold into conformations with an all-*anti* topology to give rise to a filament, ~70% of which was observed to be *βββ* (Figure 1D and Figure 3B). The resulting filaments contained two columns of Phe sidechains from the terminal residues on one of their surfaces and a single column of Phe sidechains from the residue in the middle of the peptides on the other surface. Three of these filaments were observed to bundle together to bury their surfaces with double Phe columns, leaving the single Phe columns facing outward. An additional filament was further packed against the bundle to minimize the exposure of the single Phe columns.

When peptides such as FFI, FIF, and IFF contained one Phe fewer, the observed packing modes were clearly dependent on the sequence. FIF also predominantly adopted the all-*anti* conformations, ~61% of which were *βββ*, and formed filaments with similar geometries to those of FFF except that one of its surfaces harbored a single column of Ile instead of Phe. These filaments formed bundled structures similar to those of the FFF fibers, although the resulting bundle structure was not associated with any additional filaments. On the other hand, both FFI and IFF assumed mainly an all-*syn* conformation, about 72–85% of which was *α_R_α_R_β* (Figure 3B). The filaments formed by these peptides had a convex surface containing all of the sidechains and a concave surface without any sidechain at all. A bundle structure was formed by three of these filaments so as to avoid the exposure of the Phe/Ile columns.

When the peptides contained even fewer or no Phe, the heterogeneity of the conformation and molecular packing increased. For instance, there was 53% chance of finding FII folded into the all-*syn* conformations, but the chance of finding the peptides in the other three types of conformations (all-*anti*, N-*syn*, and C-*syn*) was also significant (~40%) with the all-*anti* conformations being more favored. This heterogeneity was manifested by one of the representative simulated structures of the fibers of FII in which three filaments of peptides assumed *α_R_α_R_β* but in which one filament was *βββ* (Figure 3A). The binding interface was composed of the convex surfaces of the former three filaments and the double-columned surface of the latter filament. The conformational heterogeneity was increased in the fibers of IIF (Figure 3B), which were usually formed by two filaments with peptides in the all-*anti* conformations and another two in the all-*syn* conformations. As for III, which contains no Phe, the diverse conformations were also observed in their fibers, but the all-*anti* conformations appeared more favored. The only exception to this was IFI, which assumed predominantly all-*syn* conformations, ~81% of which were *α_R_α_R_β*, and, in order to maximize contacts between Phe sidechains, formed a three-filament bundle as did IFF and FFI.

Collectively, these results suggest that the molecular packing of tripeptides in the fibrous assemblies are linked closely to both the conformation and sequence of the tripeptides. The conformation of the peptides determines the shape of the lateral surfaces of the filaments and the orientation of the sidechains; the sequence of the peptides dictates the types of sidechain groups that could be presented on a particular surface. Although the results of the self-assembly of the eight sequences may not clearly establish rules for an accurate prediction of molecular packing in the fibrous assemblies, they revealed that, as a rule of thumb, the filaments always tend to pack laterally against each other to reduce the exposure of Phe and maximize their direct contacts.

### 2.5. Molecular Basis of Conformational Preference of Tripeptides in Self-Assembled Structures

In the determination of the molecular packing in the assembled structures, there is clearly an interplay between sequence and conformation of tripeptides. It remains unclear, however, why a particular conformation would be preferred over others in the assembled states of a tripeptide with a given sequence. To address this question, we followed our previous approach [34] and conducted the self-assembly simulations with conformational constraints (see the SI). In this type of simulation, all the peptides in the system are constrained in a given conformation, and their assembly behaviors are subsequently examined. A systematic assessment of the assembly behaviors of the same peptide in different conformations could provide insight into the conformational preference of the peptide in its assembled states.

Figure 4A shows the assembled fibrous structures of the tripeptides that were constrained in either the *βββ* conformation or the *α_R_α_R_β* conformation. The peptides in either conformational state are capable of self-assembling into fibrous structures, indicating that both conformations, regardless of amino acid sequences, were geometrically permitted to assemble the tripeptides into fibrous structures. Hence, the observed conformational preference was more likely to be attributable to the variation in stability of the different fibrous structures.

Although the calculation of free energy differences between aggregates remains difficult, we sought to gain insights in the preference for assembled conformations by systematically analyzing the noncovalent interactions present in different fibrous structures. The interactions analyzed included the van der Waals (vdW) contacts between Phe/Ile sidechains and the HB interactions between backbones. In addition, the accessibility of hydrophobic sidechains was estimated by counting the contacts between solvent particles and these sidechains.

Figure 4B shows the strength of noncovalent interactions arising in fibrous assemblies formed by each tripeptide in either the *βββ* or *α_R_α_R_β* conformation. When tripeptides such as FFF, FFI, IFF, and IFI assumed the *α_R_α_R_β* conformation in the fibrous assemblies, the Phe sidechains were more buried in the core of the assemblies than they were when the peptides assumed the *βββ* conformation, as indicated by the “PHESC-W” bars in Figure 4B. This was expected since all these peptides have a Phe in the middle of their sequences that would be exposed if the peptides folded into the *βββ* conformation. On the other hand, the peptides in *βββ* appeared to be more capable of forming HBs with their neighbors in the filaments (the “HB” bars), presumably due to a backbone geometry being more suited to the formation of interchain HBs. Thus, the data shown here indicate that the two conformations were favored by different types of noncovalent interactions, although these data may be insufficient to determine the preferred conformation. It should be noted that, when assuming a *βββ* conformation, the middle Phe residues of FFI, IFF, and IFI were more exposed than those of FFF, probably because the packing between the outward-facing Phe residues and additional filaments is only possible in the FFF fibers. Conversely, the Phe residues of FFF in the *α_R_α_R_β* structure were more exposed than those of FFI, IFF, and IFI probably because the three Phe columns on the convex surface of the filaments were too bulky to be buried fully in the cores through the lateral association between the filaments. Altogether, this result indicates that FFI and IFF were more likely than FFF to assume *α_R_α_R_β* conformation in the assemblies, which is in accord with the trend observed in the unconstrained simulations.

For the other four peptides, FIF, FII, IIF, and III, an Ile residue was in the middle of the sequences, and when these peptides assumed a *βββ* conformation, the columns composed of these Ile sidechains were generally highly exposed in the fibrous assemblies. Notwithstanding this, the unconstrained simulations indicated that FIF preferred the *βββ* conformation over *α_R_α_R_β* in the assemblies despite the exposure of the Ile columns. This is perhaps because in this conformation, the peptides presented their two Phe sidechains on a flat surface, allowing them to be fully buried through lateral association, which is not possible if the peptides assume *α_R_α_R_β* (the third column in Figure 4A). This trend of conformational preference was opposite to what was found for IFI; that is, in the assemblies, the peptides chose to assume the *α_R_α_R_β* conformation rather than the *βββ* conformation, avoiding exposing their central Phe despite the extensive interactions involving Ile that were possible when the peptides adopted a *βββ* conformation. It seems that Phe has a greater tendency than Ile to be buried in the core of the assemblies, which is consistent with the fact that Phe is more hydrophobic than Ile. For peptides such as IIF, FII, and III, on the other hand, the advantage of a *βββ*-assembled conformation diminished since there were fewer Phe sidechains that needed to be buried. This may explain why in the unconstrained simulations, the chance of finding *α_R_α_R_β* in the assembled structures became non-negligible for these three peptides.

## 3. Discussion

Precise control of the self-assembly of peptides into distinct nanostructures has been a long-standing goal for research of bioinspired materials, and the realization of this goal depends on a clear understanding of the sequence–structure relationship. The main obstacle to such understanding remains to be a lack of the means to obtain information about the atomic details of assembled structures for peptides with varying sequences. In this study, we sought to gain insight into this relationship by computationally exploring the sequence space of a tripeptide model containing phenylalanine and isoleucine using a hybrid-resolution method that allows for efficient simulations of assembly processes of peptides while retaining the capability of capturing atomic details. Our computational approach predicted that all sequences of the model tripeptide could assemble into fibrous assemblies, and this was further validated experimentally. Moreover, these fibrous assemblies were observed to consist of bundles of laterally associated filaments in which peptide chains are attached together through HBs in the direction of fiber axes, an architecture reminiscent of peptide amyloid [37]. Despite the similar assembly morphologies and hierarchies that were observed, our simulations revealed that the molecular arrangements within the fibrous assemblies were sequence specific. A systematic comparison of these observed structural details allowed us to interrogate the molecular arrangements in peptide assemblies and question how they are encoded in amino acid sequences.

A key to addressing this question lies in the determination of the conformational preference of tripeptides in the assembled states, which, in general, is challenging because short peptides like tripeptides can still have a great number of possible conformations. However, our simulations showed that only two of them, namely, *βββ* and *α_R_α_R_β*, are frequently found in fibrous assemblies. The preference for these two conformations in fibrous assemblies is perhaps because they allow tripeptides to exhibit special geometries suitable for packing in filaments. These special geometries are a parallel or antiparallel alignment between sidechain vectors and their orthogonal alignment with respect to backbone HB directions, as was pointed out previously in the study of the self-assembly of dipeptides [34]. Notably, both conformational states were found as the favorable assembly conformation for the multiple tripeptides was tested and agreed with the crystal structures of several others that had been reported previously [16,17,35,36,38], suggesting that they may be generic conformational features of tripeptides in fibrous assemblies. This finding could have an important implication in the design of assembling tripeptides: these conformations might serve as templates with which the design could begin, especially in a situation in which the design has to otherwise rely on either known experimental structures or a large number of conformational decoys, generated computationally [35]. The feasibility of this design idea awaits further investigation.

Our computational study permitted us to rationalize the connection between the sequence of peptides and their packing in fibrous assemblies formed by tripeptides. Since these assemblies arise from the lateral association of filaments, how the filaments interact with each other should be dictated by the structural properties of their lateral surfaces. Our simulations showed that the geometries of these surfaces depended on the conformations of the component tripeptides. The tripeptides in a *βββ* conformation assembled into filaments resembling typical β-sheets with sidechains placed alternatively on the two lateral sides of the filaments. When the peptides folded into an *α_R_α_R_β* conformation, the resulting filament structures were less typical, exhibiting a convex lateral surface harboring all the sidechains. These geometric features of filaments remained similar for different peptides as long as they assumed the same conformations. Nonetheless, the types and the positions of amino acid sidechains on the lateral surfaces vary, depending on sequences. Thus, both the conformational states and the sequence of tripeptides could affect the molecular packing in the assemblies.

In previous studies, experiments and extensive simulations with low-resolution models have been conducted to identify the sequence pattern that could enable tripeptides to self-assemble [15,16,23,35]. Here, we seek to address another important question at a more detailed level: why does a tripeptide with a given sequence prefer a particular type of assembled conformations and molecular packing in assemblies? We found that this preference is probably due to the need to maximize π–π interactions between Phe sidechains, van der Waals interactions between Ile sidechains, and HB interactions between backbones, and to reduce the exposure of Phe and Ile. The simulations showed that the filaments in the *α_R_α_R_β* conformation employed their convex surfaces to participate in the lateral interactions, usually leading to more hydrophobic contacts and diminished exposure of hydrophobic groups compared with those in *βββ*. The filaments in *βββ*, on the other hand, often left the sidechains of the middle residues exposed but allowed for the formation of stronger HB interactions. Hence, neither conformation was definitely favored over the other, and the conformational preference was determined by multiple competing factors. The general trend revealed by the simulations was that the favorable conformation and packing mode of tripeptides should avoid exposing the Phe sidechains and maximize their contacts as long as the peptides contained at least two Phe residues or one Phe in the middle of the sequence. Otherwise, the stability difference between the two types of conformations and packing modes may diminish, and assemblies containing both could emerge. The findings above may provide additional crucial details for a more precise design of assembling tripeptides.

## 4. Materials and Methods

### 4.1. Reagents

NH_2_- and Ac-terminated tripeptides were purchased as lyophilized powders from ChinaPeptides Co., Ltd. (Shanghai, China), at a purity level of >96%. Chemicals purchased from Sigma-Aldrich (St. Louis, MO, USA) were used without further purification. Water was processed using a Cascada BIO-water purification system (Pall Corporation, Port Washington, NY, USA) with a minimum resistivity of 18.2 MΩ⋅cm.

### 4.2. Preparation

To avoid any pre-aggregation, fresh stock solutions were prepared for all experiments by the following procedures. First, the dimethylsulfoxide (DMSO) and water were heated to target temperatures (60 ℃); then peptide stock solutions were prepared by dissolving lyophilized tripeptide powder in DMSO at a concentration of 100.0 mg/mL. The stock solutions were diluted to a desired concentration (0.2 mg/mL) in water. After standing at target temperatures (60 ℃) for 30 min and slow-cooling down to room temperature (RT), the assembly into ordered structures was observed visually.

### 4.3. Scanning Electron Microscopy

Immediately after dilution in water at a final concentration of 0.2 mg/mL, tripeptide solutions were placed on silicon slides, left to dry at 25 ℃, and then coated with platinum. Scanning electron microscopy images were made using a Zeiss SUPRA 55 (Jena, Germany) microscope operating at 20 kV. 

### 4.4. Models and Simulation Setup

The parameters used in our PACE model were obtained from our previous study [34]. All simulations in this study were conducted using the GROMACS 5.0.6 package [39]. REMD simulations of single tripeptides were performed with 12 replicas at temperatures of 310–405 K. The simulation temperature was kept constant using the Nosé–Hoover method with a coupling constant of 1 ps^−1^. Exchange between replicas was attempted every 12.5 ps, leading to an average acceptance ratio of >20%. Each replica lasted for 200 ns. The data collected at 310 K were used for analysis.

The simulation setup for each system modeled with PACE is summarized in Appendix A. In general, 30 tripeptides were initially randomly dispersed in a simulation box. MARTINI CG water was applied to solvate the system. We first performed a 5000-step initial relaxation by carrying out a steepest-descent minimization method, followed by a 2 ns simulation conducted at a temperature of 298 K with the Berendsen thermostat. Tripeptides were positionally restrained during the energy minimization. After that, we removed the constraints and conducted another 5 ns simulation at 310 K to further relax the system. The simulation time step was set to be 5 fs. Next, we performed the production run for 2–3 μs at 310 K and 1.0 atm. The Nosé−Hoover method and the Parrinello−Rahman method were used to maintain the simulation temperature and pressure, respectively.

In simulated annealing simulations, the systems were heated and annealed repeatedly for 3–5 cycles. In each cycle, the simulations started at the highest temperature for 50–200 ns and then lowered their temperature at a rate of 0.5 K/ns. After that, the systems were kept at the lowest temperature for 20 ns before another annealing cycle started over. After these annealing cycles, they were kept at 310 K for a further 2–3 μs.

Finally, in conformationally constrained simulations, all the tripeptides were constrained in the same conformational states using harmonic forces applied to backbone degrees of freedom ϕ_1_, ψ_1_, ϕ_2_, ψ_2_, ϕ_3_, and ψ_3_. The force constants and the equilibrium values of the harmonic forces were optimized against the results of the unconstrained self-assembly simulations (Appendix A). These parameters are listed in Appendix A.

## Figures and Tables

**Figure 1 ijms-21-08431-f001:**
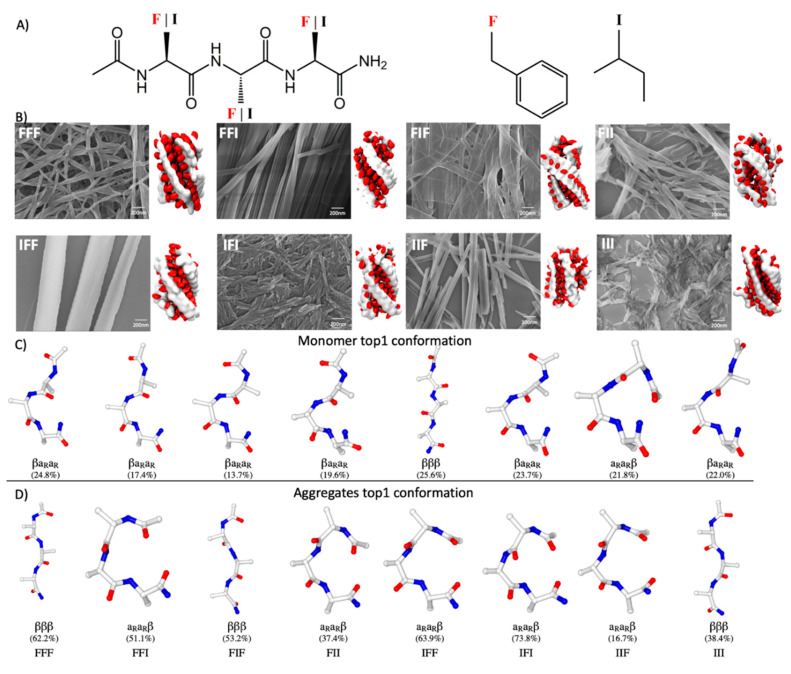
Sequence-dependent assembly structures of tripeptides containing Phe and Ile. (**A**) Schematic representation of the tripeptide sequences studied. (**B**) Assembly structures of tripeptides. Left: SEM images of tripeptide assemblies; right: a corresponding simulated assembled supramolecular structure. The backbones and sidechains of tripeptides are shown as white and red ellipsoids, respectively. Shown in (**C**,**D**) are representative structures and corresponding probabilities of the most populated conformations of tripeptides in solution states and in assembled states, respectively. For simplicity, only backbone and C_β_ atoms are shown.

**Figure 2 ijms-21-08431-f002:**
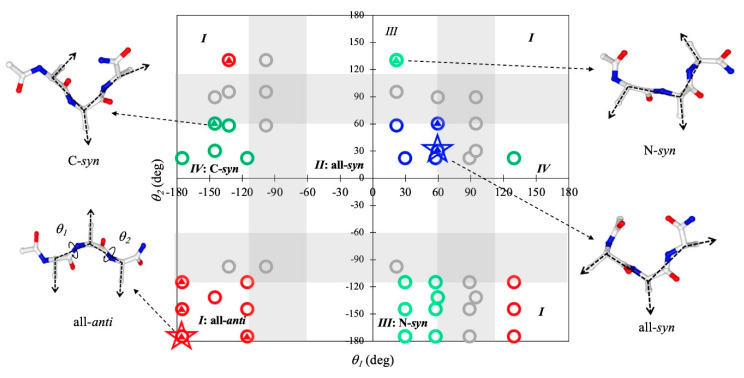
Orientations of sidechains for all the 48 possible conformational states of a tripeptide. The middle plot shows the rotational angles (*θ*_1_ and *θ*_2_) of sidechains for each conformational state. The blue, red, light green, and green circles denote all-*syn*, all-*anti*, N-*syn*, and C-*syn* conformations, respectively, and the gray circles denote the conformations with 60°<|*θ*_1_ or *θ*_2_|<115°. The conformations that were observed with a probability of >5% for at least one sequence are labeled with triangles. The *βββ* and *a_R_a_R_β* conformations are labeled with stars.

**Figure 3 ijms-21-08431-f003:**
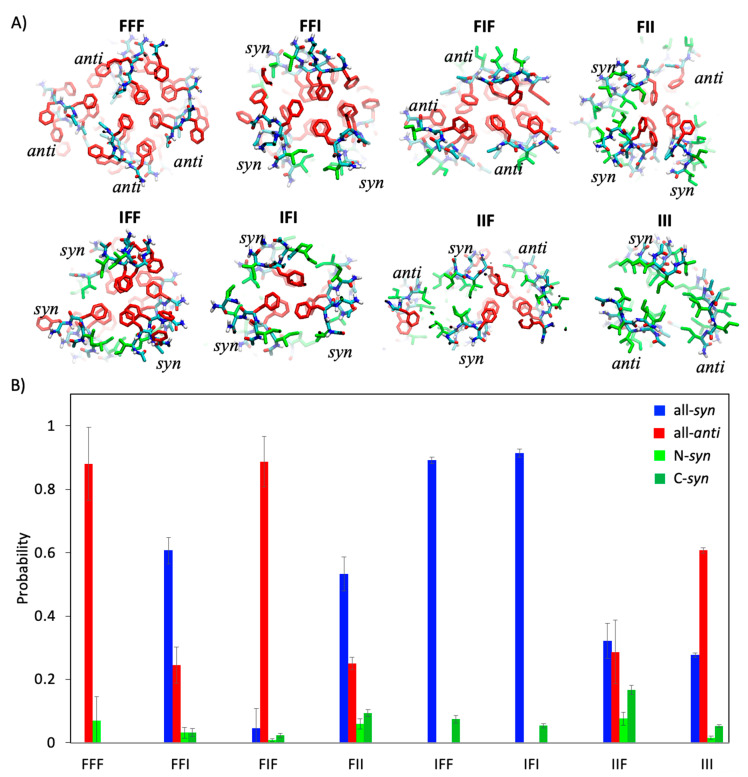
(**A**) Cross-section view of assembled structures of tripeptides. The red and green sticks denote the sidechains of Phe and Ile, respectively. (**B**) The probabilities of peptides with all-*syn* (blue bars), all-*anti* (red bars), N-*syn* (light green bars), and C-*syn* (green bars) topologies. All error bars were obtained using three independent simulations.

**Figure 4 ijms-21-08431-f004:**
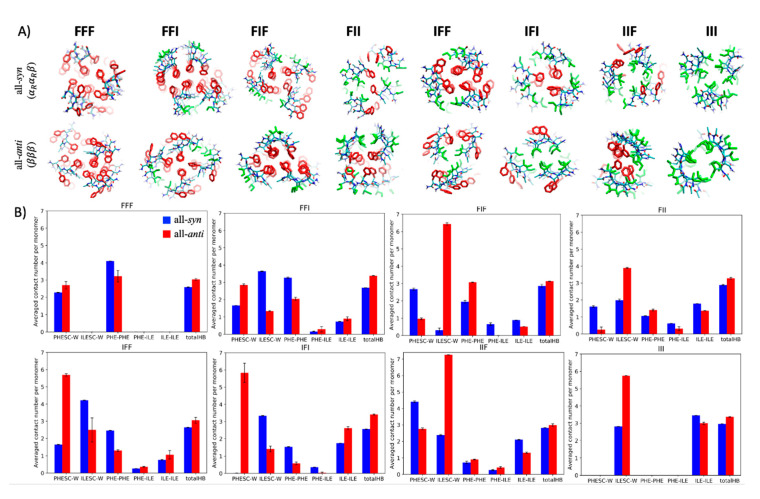
(**A**) Cross-sectional view of the assembled structures of tripeptides from conformationally constrained simulations. The first and the second rows show the results when the peptides were restrained in an *α_R_α_R_β* conformation and in a *βββ* conformation, respectively. The red and green sticks denote the sidechains of Phe and Ile, respectively. (**B**) Plot of noncovalent interactions that on average a single peptide was involved with in the assembled structures, including the sidechain–solvent interaction calculated as the water count around a Phe (“PHESC-W”) or Ile (“ILE-SC-W”) sidechain; the number of contacts between Phe sidechains (“PHESC-PHESC”), between Ile sidechains (“ILESC-ILESC”), and between sidechains of the two amino acid types (“PHESC-ILESC”); and the number of hydrogen bonds formed between peptide chains (“HB”). A contact was considered to be present if two particles were within 0.65 nm of each other. An HB was thought to arise if the distance between donor and acceptor atoms was <0.35 nm and the donor-hydrogen-acceptor angle was >120°. All the contact counts except for HBs were per-residue-based. Strictly speaking, *π*–*π* interactions involve the contacts of Phe aromatic rings that exhibit special relative orientations, but here we used the contact count (“PHESC-PHESC”) as an approximate estimate to assess the extent of the *π*–*π* interactions. Only those peptides within 1 nm of the centroid of fibrous segments were considered in the calculation. The blue and red bars denote results obtained from the simulations when the peptides were restrained in *α_R_α_R_β* or *βββ*, respectively.

**Table 1 ijms-21-08431-t001:** Similarity between conformations sampled in this study and those seen in the crystal structures of assembling tripeptides reported in previous experimental works.

	**all-*anti***			
**Sequence [ref ID]**	**DYF [16]**	**Ac-YLD [35]**	**Boc-Leu-Val-Ac12c-OMe [17]**	
Conformation states ^a^	*βββ*	*βββ*	ββα_R_	
RMSD (Å)	0.291	0.822	0.744	
	**all-*syn***			
**Sequence [ref ID]**	**Ac-LLE [36]**	**YFD [16]**	**hyp-FF [21]**	**pro-FF [21]**
Conformation states	*α_R_α_R_β*	α_L_α_L_PPII	βα_R_β	βα_R_β
RMSD (Å)	0.497	1.623	1.063	1.021

^a^ All conformations sampled were compared with the crystal structures, and for each crystal structure, only the conformation that gave the smallest root mean square distance (RMSD) was reported.

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
