# Peer review of "Sequence-Dependent Nanofiber Structures of Phenylalanine and Isoleucine Tripeptides"

_ijms, 2020, doi:10.3390/ijms21228431_

Round 1

Reviewer 1 Report

This paper presents an elegant study on molecular simulation of the self-assembly of short hydrophobic tripeptides. By using their previously implemented hybrid-resolution simulation approach, (published in ACS in 2019), the authors highlight the close interplays between the sequence of tripeptides composed only of Ile and Phe and the final architecture of the assemblies. The results of this study are important for the field and could ultimately support the rational design of peptide assemblies with tailored supramolecular structures and functionalities. The paper is well written and the scope of the paper is appropriate for IJMS. Thus, I have no hesitation in recommending publication.

I would suggest the authors to better describe the conditions of self-assembly (time, concentration, etc) in the main text, i.e. associated with Figure 1B.

In the abstract, third line, please change connection by relationships. Also, third line of the abstract, please change intermolecular structures to supramolecular structures.

Author Response

Response to Reviewer 1

We thank the review’s positive comments and suggestions. We have revised the manuscript accordingly and addressed all the raised questions as below. The changes made to the manuscript are marked in red color.

Reviewer’s question: 1) I would suggest the authors to better describe the conditions of self-assembly (time, concentration, etc) in the main text, i.e. associated with Figure 1B.

Authors’ response: We thank the reviewer for this suggestion. The conditions of our simulations have been further explained in the second paragraph on page 4:

“For each sequence, three self-assembly simulations (see Methods and Table S1), each with a length of ~3 ms, were conducted independently with the improved PACE [34]. Each simulation began with 30 tripeptides randomly distributed in a box with a size of 6 × 6 × 6 nm3 at ~100 mg/ml, a concentration higher than that used in experimental studies but comparable to those in previous computational studies of peptide self-assembly. We observed the formation of fibrous aggregates usually within the first half of microsecond of the simulations (Figure 1B), and the aggregates, once formed, remained stable till the end of the simulations”.

Reviewer’s question: 2) In the abstract, third line, please change connection by relationships. Also, third line of the abstract, please change intermolecular structures to supramolecular structures.

Authors’ response: We have made the changes accordingly.

Reviewer 2 Report

This paper reports on sequence-structure relationships in nanofibers assembled from tripeptides containing phenylalanine and isoleucine. A hybrid-resolution model termed PACE in which the solvent model is coarse-grained while the atomic details of peptides are retained is used for this purpose. The sequence-structure relationships are reasonably discussed based on the experimental results that show competing noncovalent forces. I recommend publication of this paper in International Journal of Molecular Sciences after the following points are addressed.

1) Line 141, Table S2 should be Table S3.

2) Line 147, Table S3 should be Table S4.

3) Line 156-158, the explanation for “syn” and “anti” seems opposite.

4) Line 389-390, there is a statement on the importance of π-π interactions. How the π-π interactions between phenylalanine side chains considered to be present in the PACE model? Is the PHESC-PHESC contact regarded as π-π interactions? Please clarify this point.

Author Response

Response to Reviewer 2

We appreciate the reviewer’s effort in reviewing our manuscript. We have revised the manuscript according to the reviewer’s suggestions as detailed below. All the changes made are marked in red color.

Reviewer’s question: 1) Line 141, Table S2 should be Table S3; 2) Line 147, Table S3 should be Table S4.

Authors’ response: We apologize for the errors. All tables are now correctly cited.

Reviewer’s question: 3) Line 156-158, the explanation for “syn” and “anti” seems opposite.

Authors’ response: We have corrected the errors.

Reviewer’s question: 4) Line 389-390, there is a statement on the importance of π-π interactions. How the π-π interactions between phenylalanine side chains considered to be present in the PACE model? Is the PHESC-PHESC contact regarded as π-π interactions? Please clarify this point.

Authors’ response: We thank the reviewer for pointing out this clarity issue. We have added the following statement in the caption of Figure 4 to explain how we considered the p-p interactions between Phe sidechains to arise:

“Strictly speaking, π-π interactions involve the contacts of Phe aromatic rings that exhibit special relative orientations, but we used here the contact count (“PHESC-PHESC”) as an approximate estimate to assess the extent of the π-π interactions”.